# Health-related quality of life of daily-life-affected benign essential blepharospasm: Multi-center observational study

**Parima Hirunwiwatkul**[1,2]*, **Wajamon Supawatjariyakul**[1,2], **Supharat Jariyakosol**[1,2], **Supanut Apinyawasisuk**[1,2], **Jiruth Sriratanaban**[3], **Yuda Chongpison**[4,5], **Priya Jagota**[6], **Nipat Aui-aree**[7], **Juthamat Witthayaweerasak**[7], **Suwanna Setthawatcharawanich**[8], **Kitthisak Kitthaweesin**[9], **Niphon Chirapapaisan**[10], **Piyawadee Chaimongkoltrakul**[11], **Poramaet Laowanapiban**[11], **Linda Hansapinyo**[12], **Suthida Panpitpat**[13], **Sireedhorn Kurathong**[14], **Jirat Nimworaphan**[14], **Suntaree Thitiwichienlert**[15], **Kavin Vanikieti**[16], **Narong Samipak**[17], **Worapot Srimanan**[18], **Nattapong Mekhasingharak**[19], **Pareena Chaitanuwong**[20]

1 Faculty of Medicine, Department of Ophthalmology, Chulalongkorn University, Bangkok, Thailand, 2 Ophthalmology Department, King Chulalongkorn Memorial Hospital, Bangkok, Thailand, 3 Faculty of Medicine, Preventive and Social Medicine Department, Chulalongkorn University, Bangkok, Thailand, 4 Faculty of Medicine, Center of Excellence in Biostatistics, Research Affairs, Chulalongkorn University, Bangkok, Thailand, 5 The Skin and Allergy Research Unit, Chulalongkorn University, Bangkok, Thailand, 6 Chulalongkorn Centre of Excellence for Parkinson's Disease and Related Disorders, Department of Medicine, King Chulalongkorn Memorial Hospital, Thai Red Cross Society, Bangkok, Thailand, 7 Faculty of Medicine, Department of Ophthalmology, Prince of Songkla University, Songkla, Thailand, 8 Faculty of Medicine, Department of Internal Medicine, Prince of Songkla University, Songkla, Thailand, 9 Faculty of Medicine, Department of Ophthalmology, Khon Kaen University, Khon Kaen, Thailand, 10 Faculty of Medicine Siriraj Hospital, Department of Ophthalmology, Mahidol University, Bangkok, Thailand, 11 Department of Ophthalmology, Mettapracharak (Wat Rai Khing) Hospital, Nakorn Pathom, Thailand, 12 Faculty of Medicine, Department of Ophthalmology, Chiang Mai University, Chiang Mai, Thailand, 13 Department of Ophthalmology, Udon Thani Hospital, Udon Thani, Thailand, 14 Faculty of Medicine Vajira Hospital, Department of Ophthalmology, Navamindradhiraj University, Bangkok, Thailand, 15 Faculty of Medicine, Department of Ophthalmology, Thammasat University, Pathumthani, Thailand, 16 Faculty of Medicine Ramathibodi Hospital, Department of Ophthalmology, Mahidol University, Bangkok, Thailand, 17 Department of Ophthalmology, Chakri Naruebodindra Medical Institute, Samut Prakan, Thailand, 18 Department of Ophthalmology, Phramongkutklao Hospital, Bangkok, Thailand, 19 Department of Ophthalmology, Faculty of Medicine, Naresuan University, Phitsanulok, Thailand, 20 Department of Ophthalmology, Rajavithi Hospital, Bangkok, Thailand

* hparima@gmail.com

## Abstract

### Purpose

To compare Thais' health-related quality of life (HRQOL) and severity grading, efficacy and safety in daily-life-affected benign essential blepharospasm (BEB) patients at baseline and after Botulinum toxin type A (BTX-A) treatment.

### Design

Prospective-observational study.

**Data Availability Statement:** All relevant data are within the manuscript and its Supporting Information files.

**Funding:** PH received funding form Ratchadapiseksompotch Fund, Faculty of Medicine, Chulalongkorn University, grant number RA64.2.2.2/504.1 The funders had no role in study design, data collection and analysis, decision to publish, or preparation of the manuscript.

**Competing interests:** The authors have declared that no competing interests exist.

## Participants

BEB patients with Jankovic rating scale (JRS) at least 3 in both severity and frequency graded from 14 institutes nationwide were included from August 2020 to June 2021.

## Methods

Demographic data, HRQOL evaluated by the Thai version of EQ-5D-5L and NEI-VFQ-25 questionnaires, and severity grading score evaluated by Jankovic rating scale (JRS) at baseline, 1, and 3 months after the treatment were collected. The impact of the BTX-A injections and their complications were recorded.

## Results

184 daily-life-affected BEB patients were enrolled; 159 patients (86.4%) had complete data with a mean age of 61.40±10.09 years. About 88.05% were female, and 10.1% were newly diagnosed. Most of the patients had bilateral involvement (96.9%) and 12.6% had history of BEB-related accident. After BTX-A treatment, HRQOL improved significantly in 4 dimensions of EQ-5D-5L, except self-care. The EQ_VAS (mean±SD) was 64.54±19.27, 75.13 ±15.37, 73.8±15.85 (p<0.001) and EQ-5D-5L utility score was 0.748±0.23, 0.824±0.19 and 0.807±0.19 at baseline, 1, 3 months after treatment, respectively. From NEI-VFQ-25, HRQOL also improved in all dimensions, except eye pain. The JRS improved in all patients. Self-reported minor adverse events were 22.6%, which mostly resolved within the first month.

## Conclusion

Daily-life-affected BEB impacted HRQOL in most dimensions from both generic and visual-specific questionnaires. BTX-A treatment not only decreased disease severity, but also improved quality of life.

## Introduction

Benign essential blepharospasm (BEB) is an abnormal bilateral contraction of eyelid muscles leading to episodic closure of the eyelids. The etiology of BEB is unknown with prevalence ranging from 1.6-30/100,000 worldwide and 1.6/100,000 in Thailand [1]. BEB is more commonly found in females (50–85%) [2–6] with age of onset of 40–60 years old. There is a higher degree of symptom severity and frequency of BEB in female than male [5]. Blepharospasm is one of focal dystonias whose symptoms commonly spread to other group of muscles beyond orbicularis oculi, mostly involving oromandibular and neck muscles [7–9]. Disease affects both motor and non-motor function including psychiatric problems, sleep quality, cognitive and ocular symptoms [5, 10, 11]. In severe cases, unpredicted abrupt forceful eyelid closure causes functional blindness, which may lead to life-threatening accidents while working or controlling vehicles [12, 13]. Symptoms gradually progress and could finally affect patient's physical and psychological health. Although there is no curative treatment for BEB, the standard symptomatic treatment is periodic injection of Botulinum toxin type A (BTX-A) every 3–4 months. The estimated cost is 2,000–3,000 Thai Baht (approximately 60–90 USD) per injection. Even though there have been many studies about health-related quality of life (HRQOL) in BEB patients [14–17], the use of BTX-A for BEB is not covered by Thai universal

health care coverage because there has been no previous study investigating the cost and benefits of BTX-A injection for BEB in Thai population. We evaluated HRQOL in Thai patients who had daily-life-affected BEB before and after BTX-A treatment, including efficacy and safety of the treatment. The results of the study might provide supporting evidence for amending the Nation's reimbursement policy for daily-life-affected BEB treatment.

## Methods

This is a prospective multicenter, observational study recruiting BEB patients treated with BTX-A from 14 centers across Thailand from August 2020 to June 2021. The study protocol was approved by the Central Research Ethics Committee (Certificate number: COA--CREC070/2020), and an Institutional Review Board of each site. The study protocol adhered to the tenets of the Declaration of Helsinki. The inclusion criteria were daily-life-affected BEB patients who had Jankovic rating scale (JRS) of at least 3 in both severity and frequency scores (JRS ≥ 6), graded by a neuro-ophthalmologist or neurologist and required BTX-A treatment. We diagnosed blepharospasm by excessive contractions of muscles around the eyes, especially orbicularis oculi. The spasm is episodic, involuntary, and unpredictable [18].

We use the JRS, which is widely used for clinical research and is easy to use [19], to assess the severity and frequency of blepharospasm. The score ranges from 0–4 in each severity and frequency rating scales. Grade 0 is no spasm. Grade 1 and 2 are defined by increased blinking or eyelid fluttering without functional disabilities. Grade 3 and 4 are defined by increased severity and frequency that disturb normal function of daily activities [20]. Then we used at least 3 in both severity and frequency scores as the inclusion criteria in this study.

We excluded patients who (1) received BTX-A treatment less than 12 weeks before enrollment, (2) aged under 18 years old, (3) were pregnant, (4) had other neurological problems or other serious physical problems, and (5) could not understand Thai language. In each center, all patients were invited to join the study by trained research assistants or treating physicians with a standard invitation script. All participants received BTX-A treatment from each center to control the spasm, and the dose depended on the severity, previous response and type of BTX-A.

Demographic and clinical data collection included: age, gender, involved eye(s), underlying disease, history of BTX-A use, current medication, previous BEB related surgery, and history of BEB related accident. Socioeconomic data included marital status, education level, occupation or employment status, income, transportation cost per healthcare visit, and health insurance.

HRQOL was evaluated by 2 questionnaires covering both vision and general health status. First, a Thai version of the 25-item National Eye Institute visual function questionnaire (NEI VFQ-25) was used. This visual specific questionnaire covered 11 subscales including: 1) general vision, 2) ocular pain, 3) difficulty with near-vision activities, 4) difficulty with distance-vision activities, 5) limitation of social functioning due to vision, 6) mental health problems due to vision, 7) role limitations due to vision, 8) dependency on others due to vision, 9) driving difficulties, 10) difficulty with color vision, and 11) difficulty with peripheral vision [21, 22]. The composite score from NEIVFQ-25 was analyzed as the mean score of all items except for the general health item. The general health was evaluated by overall health status, depends on the patients' perception. The second questionnaire was the Thai version of the European Quality of Life Five Dimension Five levels (EQ-5D-5L) which related to general health questions comprising five different dimensions including: 1) mobility, 2) self-care, 3) usual activities, 4) pain/discomfort and 5) anxiety/depression [23, 24]. Both versions of the Thai questionnaires were validated in previous studies [22, 24].

We evaluated the JRS, NEI-VFQ-25, EQ-5D-5L at baseline before the treatment, 4 to 6 weeks (the maximal effect of BTX-A), and 12 to 16 weeks after the treatment (the end of BTX-A's duration of action). In order to collect NEI-VFQ-25, EQ-5D-5L data, participants were asked to complete a case record form. Trained research assistants assisted illiterate participants to complete the form. JRS (both severity and frequency scores) and treatment complications were evaluated and recorded by the same physician for each participant throughout the 3 visits. Complications and their durations were recorded at the second and final visit. Type and amount of BTX-A, site of injection, and other prescribed medications were also documented. During the COVID-19 pandemic, some of the follow-up visits were performed via video conferencing by a single physician for each participant.

## Statistical analysis

Sample size was calculated by using data from the validated Thai NEI-VFQ-25 questionnaires study [14]. To achieve power of 80% at significant level of 0.05, the calculated sample size was 146 patients with 10% loss to follow up. Thus, we aimed to include 165 patients.

Descriptive statistics were used to describe patients' characteristics including demographic data, clinical data, and complication of BTX-A injection. Box plot over time and linear mixed model analysis were used to describe and compare HRQOL and JRS pre and post treatment, respectively. NEI-VFQ-25 and EQ-5D-5L were analyzed and adjusted with JRS summarized scores. We tested the interaction between each QoL domain and JRS with a significance level of 0.05. When significant, the interaction term is included in the reported model. Employment, sex, age, and underlying disease were evaluated as confounding factors. To determine efficacy, durations of action of BTX-A were compared by types of BTX-A using Kaplan-Meier estimates.

## Results

### Demographic data

A total of 184 participants were enrolled. One hundred and fifty-nine participants (86.41%) had complete data until the final follow-up visit. In COVID-19 endemic situation, 25 patients were loss to follow up and unable to be contacted by phone during the follow-up period: 11 patients on 1st month and 14 patients on 3rd month after treatment visit. There was no significant difference in age, sex, presence of comorbid diseases, and JRS severity grading between patients with complete data and those who were loss to follow-up. Demographic data are showed in Table 1. Participants who had complete follow-up data had a mean age of 61.40 years and were female predominate. The average duration of BEB was 5.05 ± 4.25 years and 96.86% of BEB was bilateral. Approximately half of the participants (48.43%) had prior BTX-A treatment of more than 10 times. A longer duration of the disease was found in females (5.3 years) than males (3.02 years). The minority of participants (10.06%) were newly treated with BTX-A. The majority of participants (71.70%) had comorbid diseases. Hypertension, dyslipidemia and diabetes mellitus were the three most common conditions. The JRS at baseline was 3.41 (SD 0.49, range 3–4) for severity, 3.40 (SD 0.49, range 3–4) for frequency, and 6.82 (SD 0.88, range 6–8) for summative score. There are 12.58% of participants had history of BEB-related accidents consisting of motor vehicle accidents, which was the most common accident type (68.42%), followed by falls and bumping into others. The employment rate in the complete-to-follow-up group was about two times higher than in the loss-to-follow-up group (60.38% versus 32.00%, respectively).

**Table 1. Demographic data of patients with daily-life-affected BEB at baseline.**

| General | Total Included N = 184 | Complete follow up N = 159 | Lost-to-follow up N = 25 |
|---|---|---|---|
| Age (years) (Mean, SD) | 61.57 (10.47) | 61.40 (10.09) | 62.68 (12.81) |
| Sex (Female) | 87.50% | 88.05% | 84.00% |
| Presence of U/D | 71.74% | 71.70% | 72.00% |
| DM | 13.59% | 15.09% | 4.00% |
| HT | 38.04% | 36.48% | 48.00% |
| DLP | 32.61% | 32.70% | 32.00% |
| Old CVA | 1.09% | 0.63% | 4.00% |
| Thyroid diseases | 2.17% | 1.89% | 4.00% |
| CAD | 1.63% | 1.89% | 0% |
| CKD | 1.63% | 1.26% | 4.00% |
| Other | 25.00% | 26.42% | 16.00% |
| Number of previous injections | | | |
| 0 | 11.48% | 11.95% | 8.33% |
| 1–5 | 20.77% | 21.38% | 16.67% |
| 6–10 | 17.49% | 18.24% | 12.50% |
| 11–20 | 29.51% | 28.30% | 37.50% |
| >20 | 20.77% | 20.13% | 25.00% |
| Jankovic Rating Scale; JRS (Mean, SD) | | | |
| Frequency grading | 3.41 (0.49) | 3.40 (0.49) | 3.52 (0.51) |
| Severity grading | 3.40 (0.49) | 3.41 (0.49) | 3.36 (0.49) |
| Total grading | 6.82 (0.88) | 6.81 (0.87) | 6.88 (0.93) |
| Presence of BEB-related accident | 11.96% | 12.58% | 8.00% |
| Presence of other med. | 21.20% | 18.87% | 36.00% |
| New case | 9.78% | 10.06% | 8.00% |
| Employment* | 56.52% | 60.38% | 32.00% |
| Total income* (THB/month) (Med (Q2, Q3)) | 6,000 (600, 23,150) | 8,000 (600, 30,000) | 700 (0, 8,000) |

*p-value < 0.01 for comparisons between complete follow up and lost-to-follow up groups.

CAD = Coronary artery disease, CKD = Chronic kidney disease, CVA = Cerebrovascular accident, DLP = Dyslipidemia, DM = Diabetes mellitus, HT = Hypertension, med. = medication, THB = Thai baht, U/D = Underlying diseases.

### Effect of botulinum toxin treatment on clinical outcome

The results showed a significant improvement in JRS summary scores after BTX-A treatment. Means ± SD of JRS summary scores were 6.81 ± 0.87, 1.46 ± 1.75, 5.20 ± 2.06 at 0, 1, 3 months, respectively (Fig 1). The predicted JRS scores using linear mixed model were 6.72 (95%CI 6.48–6.95, p < 0.001), 1.41 (95%CI 1.16–1.66, p < 0.001), 5.18 (95%CI 4.92–5.43, p < 0.001) at 0, 1, 3 months, respectively (Fig 2).

### Effect of Botulinum toxin treatment on HRQOL

EQ-5D-5L and NEI-VFQ-25 improved at 1 month and did not decline to pre-treatment level after 3 months in all dimensions except for self-care and ocular pain (Fig 1). EQ-5D-5L utility scores were 0.75 ± 0.23, 0.82 ± 0.19, 0.81 ± 0.19 at baseline, post-treatment 1, and 3 months, respectively; and predicted values were 0.74 (95%CI 0.72–0.77, p < 0.001), 0.83 (95%CI 0.80–0.86, p < 0.001), and 0.81 (95%CI 0.78–0.84, p < 0.001) respectively. NEI-VFQ–25 composite scores were 58.70 ± 19.07, 68.79 ± 17.37, and 66.52 ± 18.71, respectively, and predicted values were 59.53 (95%CI 56.95–62.10, p < 0.001), 69.54 (95%CI 66.85–72.23, p < 0.001), and 67.15

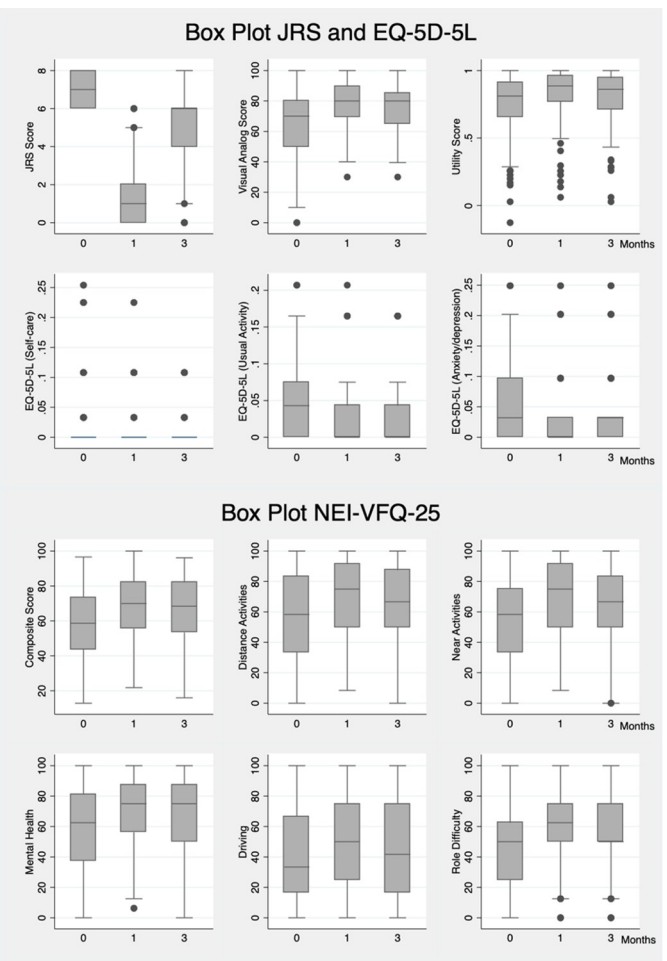

**Fig 1. Box plot JRS, EQ-5D-5L, and NEI-VFQ-25 at each visit.**

(95%CI 64.43–69.88, p < 0.001), respectively. The most significant improvement in the NEI-VFQ-25 subscale were role limitations, difficulty with near-vision activities, mental health problems, and limitation of social functioning with differing scores between pre and post-treatment at 1 month of 16.78 (95%CI 13.08–20.48, p < 0.001), 13.81 (95%CI 10.06–17.57, p < 0.001), 13.79 (95%CI 10.04–17.54, p < 0.001), and 11.80 (95%CI 7.81–15.79, p < 0.001), respectively. By subgroup analysis, pre-treatment HRQOL scores were lower in female, unemployed participants with underlying diseases and in participants with higher JRS severity scores.

After testing and finding interaction terms between HRQOL, JRS, and time, we used covariate JRS summary scores with linear mixed model. In JRS = 8 group, the utility scores (EQ-5D-5L) were 0.68 (95%CI 0.64–0.72), 0.75 (95%CI 0.66–0.85), and 0.78 (95%CI 0.73–0.82) (p-value < 0.001) at baseline, 1 month, and 3 months, respectively. There was better improvement in utility and composite scores in higher JRS summary scores than lower scores (Fig 3).

Our study selected OnabotulinumtoxinA (Botox®, Allergan) or Abobotulinum toxin A (Dysport®, Galderma Laboratories, LP) which varied among institutes based on physician's preference. The average dose of OnabotulinumtoxinA was 13.11 units per eye per treatment and the average dose of AbobolutinumtoxinA was 39.41 units/eye/treatment. Duration of

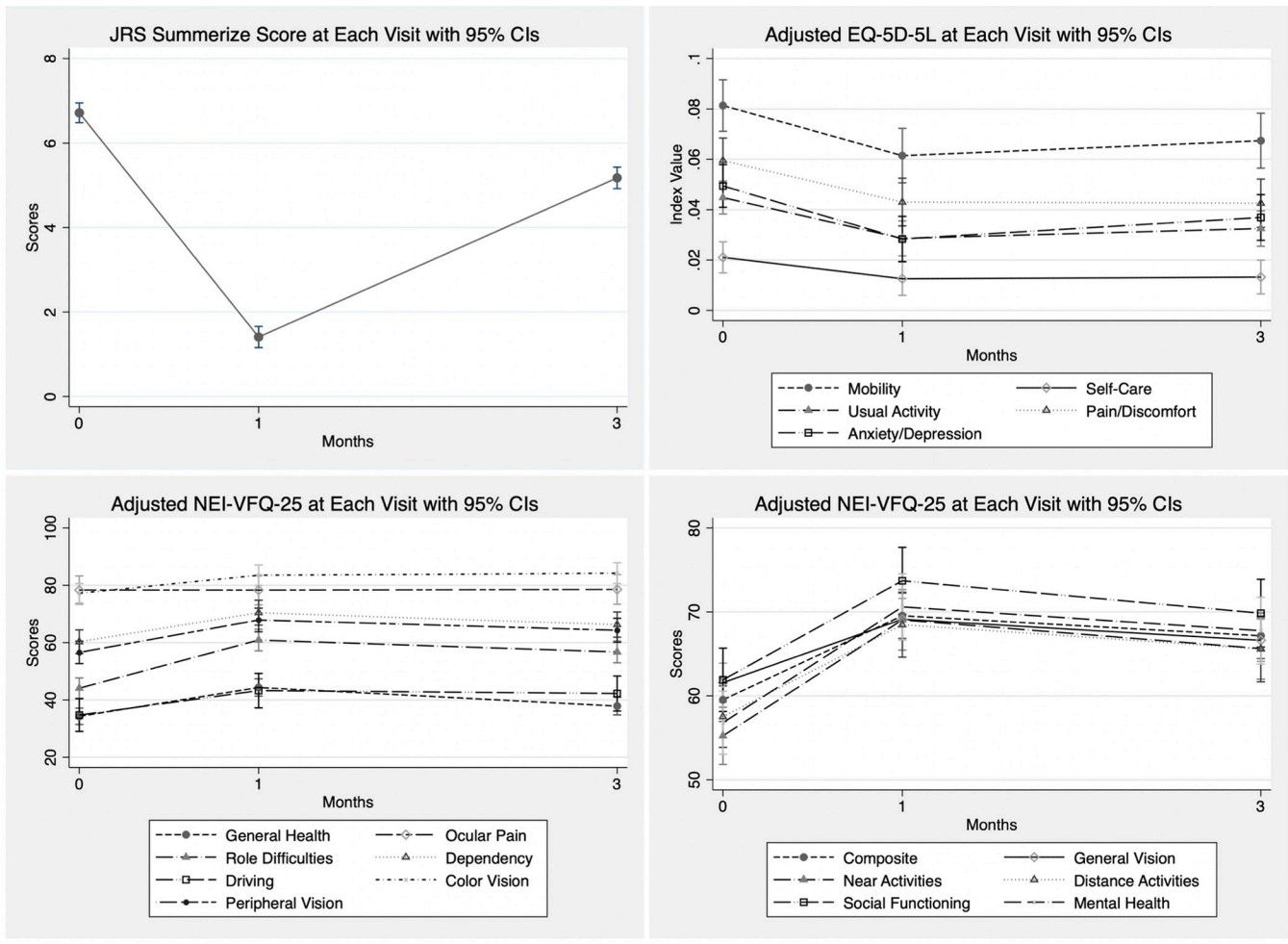

**Fig 2. JRS, EQ-5D-5L, and NEI-VFQ-25 at each visit with 95% confidence interval.** All models were adjusted for employment, sex, age, and underlying disease.

action is defined as time from BTX-A injection to the time participants started to feel eyelid spasm. We found that 50 percent of participants reported that they started to feel the beginning of blepharospasm at 10th week after treatment (i.e., median duration of action is approximately 10 weeks). Almost all participants had recurring symptoms at the end of the 12th week after treatment. There was no significant difference in duration of action between the two types of BTX-A (Log-rank test, p >0.99) (Fig 4). About 22% (36 participants) reported minor complications; ptosis (10.69%) and lagophthalmos (5.66%) were the first and second most common complications. Others included diplopia (2.52%), lips ptosis (1.26%), and others complication (5.03%) which included: dry eyes, lid edema, dizziness and neck pain. All complications were completely resolved within 4 weeks after BTX-A treatment.

## Discussion

We started our study in 2020 when the new diagnostic criteria [25] have not been published. We thus used our criteria which was an involuntary spasm of the orbicularis oculi with sensory trick without abnormal contraction of the lower face. With this inclusion criteria, we recruited 5 unilateral blepharospasm in our studied population. All 5 participants had onset of 3 years or

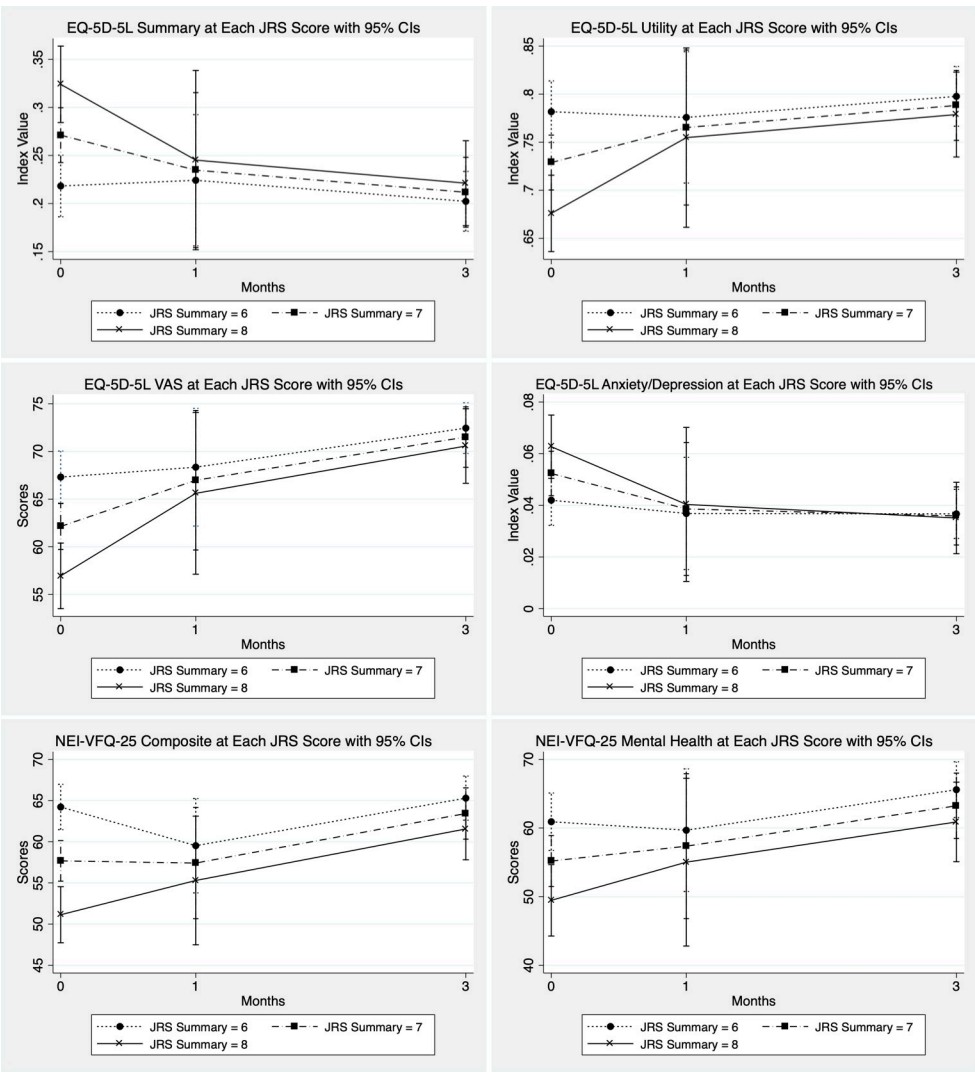

**Fig 3. EQ-5D-5L, and NEI-VFQ-25 at each JRS scores with 95% confidence interval.** Estimated QoL were from models with an interaction term between JRS and time.

less at the time of inclusion and possibly progressed to bilateral blepharospasm or hemifacial spasm in the future. Grandas et al. reported that 19.7% of the participants in their study had unilateral involvement, which mostly progressed to bilateral blepharospasm [26]. BEB had approximately 50% risk of symptoms spreading to other muscles beyond orbicularis oculi in 5 years [7, 8].

This condition can cause accidents, injury, and loss of health. In our study, accidents associated with suddenly severe blepharospasm occurred while operating motor vehicles, walking and bumping into others, and falling while walking. These accidents lead to injury and limited daily activities, especially hindering driving and working abilities. Employment or job duties were key factors in seeking BTX-A treatment. Unemployment and low income were major factors in the loss to follow-up group. Co-medication was used more in the loss to follow-up group for relief symptoms than the BTX-A treated patients. This study revealed the female-to-male ratio was higher than in previous studies [3, 27, 28]. Since our project recruited participants with high-severity grading score, 88% of our patients were female because our study

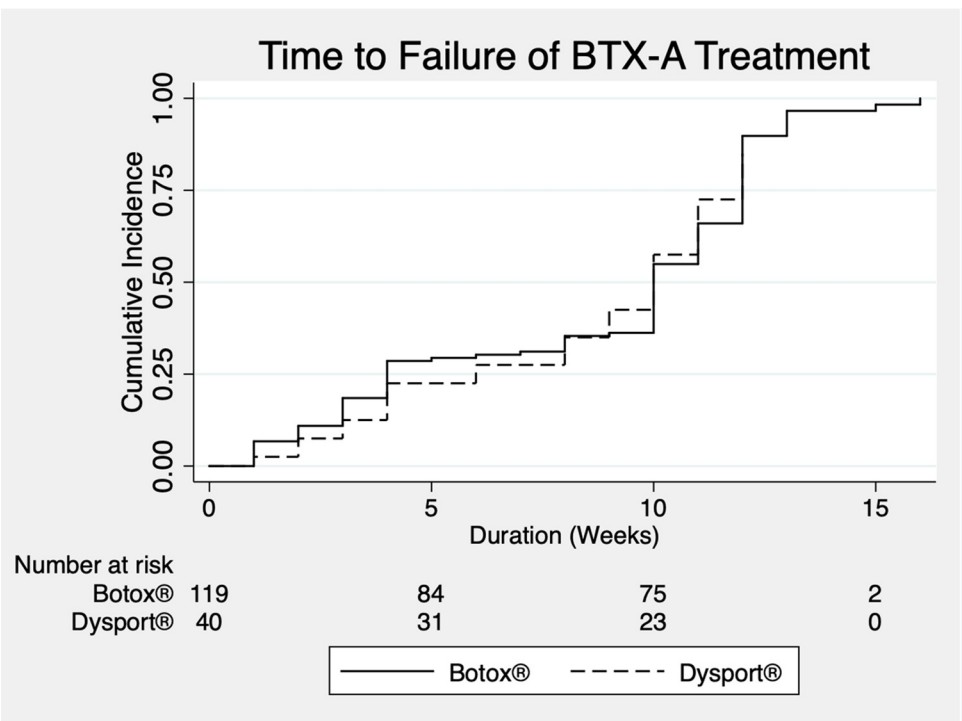

**Fig 4. Kaplan-Meier estimates of the duration of action of BTX-A treatment.**

included only severe cases who need botulinum toxin treatment. The prevalence of female in daily-life-affected BEB are mostly the same in 14 nationwide institutes of Thailand, with median prevalence of 90% (83.92%, 95%). In general, the female preponderance in BEB is approximately 60–71% [4, 27–30]. Evidence showed higher degree of symptom severity and frequency of BEB in female than in male [31, 32]. The use of Botulinum toxin A in BEB treatment was associated with the efficacy and cost of botulinum toxin A treatment. Male and female patients may perceive the affected quality of life differently, even if the severity is the same. Female patients might be more concerned about their symptoms and thus seek medical attention more than male patients.

We chose the NEI-VFQ25 and EQ5D5L questionnaires to evaluate HRQOL in the Thai population because we would like to determine the effect of DL-BEB on vision abilities and general health to assess utility score to compare the health-related effect with other diseases. In pre-treatment DL-BEB, vision effects of HRQOL measured by the NEI-VFQ-25 questionnaires were decreased to lower than 50% in role limitation, driving difficulties and general health. Ocular pain and difficulties with color vision have less effect (about 20% decrease). In Europeans, ocular pain has significant effect after BTX-A treatment [15, 33]. However, in our study, it had no effect because our patient's baseline score was higher, and there was no statistical difference in the mean scores for ocular pain between the baseline and the after-treatment measurement in our study. From NEI-VFQ25, mental health, difficulty with near-vision activities, distance-vision activities and peripheral vision had effects in the 50–60% range. Although vision itself is unaffected by BEB, spasms, repositioning and movement of eyelids during tonic-clonic spasms can disrupt vision. Hall et al. used the same BEB questionnaires in hemifacial spasm patients and found similar affected role limitation, mental health, and color vision as in our study. The general health and composite score in Thai DL-BEB was less than the US

DL-BEB (34.39, 59.53% in Thais and 54, 70% in US, respectively) [15]. From the generic HRQOL by EQ-5D-5L, utility index at baseline was 0.74 (VAS 63.60%) which improved in the optimal period of BTX-A (UI 0.83, VAS 74.84%) and dropped at 3 months after treatment. In a previous study in Germans, HRQOL had the same pattern. However, the baseline utility index was lower than in Thais [34]. It seems that quality of life is affected in the same way, but the degree of decline in HRQOL varies by race, culture and the patient's environment.

Blepharospasm affected mental health and socializing, which has been reported in many previous studies [15, 35–37]. However, our study found that moderate to severe blepharospasm which is affected daily-life activities, also decreased QoL in visual function and general health. BTX-A treatment not only reduces physical severity but also enhances quality of life, including in non-physical aspects. The higher severity grading at baseline, the more HRQOL improvement after BTX-A treatment.

The strength of this study is that we conducted a prospective multicenter, observational study from neuro-ophthalmologists and neurologists in 14 centers nationwide with large population that included participants from all parts of Thailand. Data on quality of life, collected through generic and visual-specific questionnaires, broadened the scope of data collection. However, this study did have some limitations. BEB is mostly bilateral and unpredictable. A recent multicenter study has proposed new diagnostic criteria for BEB which include 4 items: stereotyped, bilateral, and synchronous orbicularis oculi muscle spasms inducing eyelids closure/ narrowing as item 1, effective sensory trick as item 2, increased blinking as item 3, and inability to voluntarily suppress the spasms as item 4. Item 1 yields high sensitivity (96%) and relatively low specificity (76%). Combinations of more than 1 item increase sensitivity and specificity [25]. If our inclusion criteria for DL-BEB differ from these new criteria, we may include a few unilateral cases that could be misdiagnosed by the new criteria. The new patients who self-pay their medical expenses might deny Botulinum toxin treatment due to financial limitation. They, thus, were not including into the study. This resulted in another potential selection bias. This study was conducted on a short-term basis because data for only one cycle of BTX-A treatment was collected. Future study warrants a follow up of these cases to identify progression and long-term effects. Another limitation was that this study was conducted during the COVID-19 pandemic, which may have impacted the HRQOL [38–41]. The peak of the COVID-19 outbreak occurred mostly during the follow up visit period of 1 or 3 months. This timing could imply that the actual HRQOL after treatment might be higher than reported in our results.

## Conclusion

Daily-life-affected BEB treated with BTX-A significantly improved disease severity and HRQOL in most dimensions in both generic and visual-specific questionnaires, with the exception of the self-care dimension in the EQ-5D-5L and ocular pain subscale in the NEI-VFQ-25. The factors that affected QoL improvement after BTX-A treatment were gender, age, underlying diseases and employment. BTX-A treatment not only decreased disease severity, but also improved QoL.

## Supporting information

**S1 Table. Baseline quality of life by generic (EQ-5D-5L) and condition-specific (NEI-VFQ25) questionnaires (Mean, SD).**
(DOCX)

**S2 Table. Daily-life-affected BEB patients' health-related quality of life by EQ-5D-5L questionnaires in subscales.**
(DOCX)

**S3 Table. Estimated EQ-5D-5L, NEI-VFQ-25 and JRS at each visit and coefficient with 95% confidential interval of patients with daily-life-affected blepharospasm.**
(DOCX)

## Acknowledgments

We would like to express our gratitude to the Thai Neuro-ophthalmology Society (TNOS) for their team support. The EuroQol Foundation and The Health Intervention and Technology Assessment Program (HITAP) for their permission to use the Thai version of EQ5D5L questionnaires, Associate Professor Suwanna Setthawatcharawanich for her permission to use the Thai version of National Eye Institute Visual Functioning Questionnaire (NEI-VFQ25) [14, 22] in this study. Special thanks to Mr. Phanupong Phutrakool who advised the use of the REDCap program for data collection from multi-centers.

## Author Contributions

**Conceptualization:** Parima Hirunwiwatkul, Wajamon Supawatjariyakul, Supharat Jariyakosol, Supanut Apinyawasisuk, Jiruth Sriratanaban.

**Data curation:** Parima Hirunwiwatkul, Wajamon Supawatjariyakul, Yuda Chongpison.

**Formal analysis:** Parima Hirunwiwatkul, Wajamon Supawatjariyakul, Yuda Chongpison.

**Funding acquisition:** Parima Hirunwiwatkul.

**Investigation:** Parima Hirunwiwatkul, Supharat Jariyakosol, Supanut Apinyawasisuk, Priya Jagota, Nipat Aui-aree, Juthamat Witthayaweerasak, Suwanna Setthawatcharawanich, Kitthisak Kitthaweesin, Niphon Chirapapaisan, Piyawadee Chaimongkoltrakul, Poramaet Laowanapiban, Linda Hansapinyo, Suthida Panpitpat, Sireedhorn Kurathong, Jirat Nimworaphan, Suntaree Thitiwichienlert, Kavin Vanikieti, Narong Samipak, Worapot Srimanan, Nattapong Mekhasingharak, Pareena Chaitanuwong.

**Methodology:** Parima Hirunwiwatkul, Wajamon Supawatjariyakul, Supharat Jariyakosol, Supanut Apinyawasisuk, Jiruth Sriratanaban, Priya Jagota, Nipat Aui-aree, Juthamat Witthayaweerasak, Suwanna Setthawatcharawanich, Kitthisak Kitthaweesin, Niphon Chirapapaisan, Piyawadee Chaimongkoltrakul, Poramaet Laowanapiban, Linda Hansapinyo, Suthida Panpitpat, Sireedhorn Kurathong, Jirat Nimworaphan, Suntaree Thitiwichienlert, Kavin Vanikieti, Narong Samipak, Worapot Srimanan, Nattapong Mekhasingharak, Pareena Chaitanuwong.

**Project administration:** Parima Hirunwiwatkul, Juthamat Witthayaweerasak, Suwanna Setthawatcharawanich, Kitthisak Kitthaweesin, Niphon Chirapapaisan, Poramaet Laowanapiban, Linda Hansapinyo, Suthida Panpitpat, Sireedhorn Kurathong, Suntaree Thitiwichienlert, Kavin Vanikieti, Worapot Srimanan, Nattapong Mekhasingharak, Pareena Chaitanuwong.

**Resources:** Parima Hirunwiwatkul, Suwanna Setthawatcharawanich.

**Software:** Parima Hirunwiwatkul, Yuda Chongpison.

**Supervision:** Parima Hirunwiwatkul.

**Validation:** Parima Hirunwiwatkul, Wajamon Supawatjariyakul.

**Visualization:** Parima Hirunwiwatkul, Wajamon Supawatjariyakul, Supharat Jariyakosol, Supanut Apinyawasisuk, Jiruth Sriratanaban, Juthamat Witthayaweerasak, Sireedhorn Kurathong, Nattapong Mekhasingharak.

**Writing – original draft:** Wajamon Supawatjariyakul, Yuda Chongpison.

**Writing – review & editing:** Parima Hirunwiwatkul, Supharat Jariyakosol, Supanut Apinyawasisuk, Jiruth Sriratanaban, Yuda Chongpison, Priya Jagota, Nipat Aui-aree, Juthamat Witthayaweerasak, Suwanna Setthawatcharawanich, Kitthisak Kitthaweesin, Niphon Chirapapaisan, Piyawadee Chaimongkoltrakul, Poramaet Laowanapiban, Linda Hansapinyo, Suthida Panpitpat, Sireedhorn Kurathong, Jirat Nimworaphan, Suntaree Thitiwichienlert, Kavin Vanikieti, Narong Samipak, Worapot Srimanan, Nattapong Mekhasingharak, Pareena Chaitanuwong.

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
