## [Decision Letter · Decision Letter 0]

16 Nov 2022

PONE-D-22-27465Health-related quality of life of daily-life-affected benign essential blepharospasm: multi-center observational studyPLOS ONE

Dear Dr. Hirunwiwatkul,

Thank you for submitting your manuscript to PLOS ONE. After careful consideration, we feel that it has merit but does not fully meet PLOS ONE’s publication criteria as it currently stands. Therefore, we invite you to submit a revised version of the manuscript that addresses the points raised during the review process.

We look forward to receiving your revised manuscript.

Kind regards,

Tommaso Martino, M.D.

Academic Editor

PLOS ONE

Journal Requirements:

Reviewers' comments:

Reviewer's Responses to Questions

**Comments to the Author**

1. Is the manuscript technically sound, and do the data support the conclusions?

Reviewer #1: Yes

Reviewer #2: Partly

Reviewer #3: Yes

2. Has the statistical analysis been performed appropriately and rigorously? 

Reviewer #1: No

Reviewer #2: Yes

Reviewer #3: Yes

3. Have the authors made all data underlying the findings in their manuscript fully available?

Reviewer #1: Yes

Reviewer #2: Yes

Reviewer #3: Yes

4. Is the manuscript presented in an intelligible fashion and written in standard English?

Reviewer #1: Yes

Reviewer #2: Yes

Reviewer #3: Yes

5. Review Comments to the Author

Reviewer #1: The authors reported an interesting study about quality of like in patients with benign essential blepharospasm (BEB). I have some comments to the authors:

- In the methods the authors should specify that the diagnosis of BEB was made in accordance with the recent criteria. Here the paper that should be included:

Defazio G, et al. Diagnostic criteria for blepharospasm: A multicenter international study. Parkinsonism Relat Disord. 2021 Oct;91:109-114. doi: 10.1016/j.parkreldis.2021.09.004. Epub 2021 Sep 8. PMID: 34583301; PMCID: PMC9048224.

- Please specify whether all the patients had focal dystonia. The phenomenon of spread in BEB is a frequent condition, so it would be better if the authors expanded this point in the methods. There are several studies on the topic that might be helpful:

Berman BD, et al. Risk of spread in adult-onset isolated focal dystonia: a prospective international cohort study. J Neurol Neurosurg Psychiatry. 2020 Mar;91(3):314-320. doi: 10.1136/jnnp-2019-321794. Epub 2019 Dec 17. PMID: 31848221; PMCID: PMC7024047.

Ercoli T, et al. Spread of segmental/multifocal idiopathic adult-onset dystonia to a third body site. Parkinsonism Relat Disord. 2021 Jun;87:70-74. doi: 10.1016/j.parkreldis.2021.04.022. Epub 2021 May 12. PMID: 33991781.

- Among the strengths of the study, the author may also add that the Jankovic rating scale is one of the best tool to assess BEB, as recently reported in this recent international study:

Defazio G, et al. Measurement Properties of Clinical Scales Rating the Severity of Blepharospasm: A Multicenter Observational Study. Mov Disord Clin Pract. 2022 Aug 15;9(7):949-955. doi: 10.1002/mdc3.13530. PMID: 36247913; PMCID: PMC9547140.

- The results should better organize adding the p value in the text and the findings from the statistic.

- “In our study, the female to male ratio was higher than previous studies.” Please expand this point.

Reviewer #2: The authors present data about quality of life of Thai patients with blepharospasm treated with

Botulinum toxin type A. The manuscript might have important clinical message and may be of great

interest to the readers who are interested in this specific topic.

However, I do believe that the manuscript would benefit from a second round of major revisions in order to

improve it before being considered for publication. Below are some recommendations:

Q1. Line 63: “Most of the patients had bilateral involvement (96.9%)”

Please discuss why you consider few patients with unilateral involvement as having blepharospasm rather

than hemifacial spasm taking into account that all more accepted definitions describe blepharospasm as a

bilateral disturb.

Q2. Line 97-98 : “We evaluated 98 HRQOL in Thai patients who had daily-life-affected BEB before and

after…”

What do you mean by “daily-life-affected” when you consider a disease with continuous symtpoms like

blepharospasm? Do you refer to more severe forms of BEB? Please specify

Q3. Lines 107-108 “Inclusion criteria were daily-life-affected BEB patients who had Jankovic rating scale

(JRS) of at least 3 in both severity and frequency scores graded…”

Please provide a brief sentence on the scoring graduation of the Jankovic rating scale (for istance : The JRS

ranges from 0 to 8 points (sum score) and includes 2 categories:… etc etc). Moreover, a score of at least 3

for both severity and frequency on the Jankovic rating scale is relatively high as compared to those present

in several case series from the literature (see works by Jankovic J, et al. 10.1002/mds.22368; and Jankovic

J, et al. 10.1002/mds.23658), where Mean JRS sum score was less than 6 despite of a disease duration of

more than 6 years (5.05 ± 4.25 years in the present study). It would be of great interest to report which

rate of patients from the 14 centers across Thailand reached a Jankovic rating scale total score more than 6.

Had all patients higher severity scores? why was the severity of the blepharospasm in this study population

higher than that of the general population (as you stated – line 243)? The blepharospasm is a disorder

almost exclusively managed in tertiary centers worldwide and this should not be a reason why the severity

scores are so high in BEB patients in this study. Consider this in the discussion.

Q.4 Line 130: “Both versions of the Thai questionnaires were validated in previous studies”.

please report reference of the previous studies were a the Thai version of the 25-item National Eye

Institute visual function questionnaire (NEI VFQ-25) was validated.

Q5. Line 179: “Table 1: Demographic data of patients with daily-life-affected BEB”

Please specify in the table that JRS scores refer to baseline state.

Q6. Lines 221-222-223 : “Duration of action analyzed by Kaplan-Meier estimates, 50% of participants

reported the beginning of recurrent BEB symptoms at 10 weeks after the treatment.”

Please give sense to the first part of the sentence.

Q7. the General health item and composite score of the 25-item National Eye Institute visual function

questionnaire (NEI VFQ-25) are first described in the results and discussion. It would be better to introduce

them in the methods firstly. In particular, unlike the other scores, Composite scores increase after

treatment, so it should be better defined in methods section.

Q8. Figures:

most Y-X graphs’ legends are almost unreadable. Please increase graphs’ quality or limit the number of

graphs/figures in order to increase size (fig. 4 is a good solution)

Q9. Statistical analysis and results:

It would be more appropriate to specify coefficient and significance values when comparing pre and posttreatment data (both in text and tables)

Reviewer #3: This research is an accurate description of demographic and clinical features of BEB in Thailand. Authors show with adeguate statistical tools how quality of life improves after treatment with botulinum toxin. Authors used different and validated scales with reliable results.

6. PLOS authors have the option to publish the peer review history of their article (what does this mean?). If published, this will include your full peer review and any attached files.

Reviewer #1: No

Reviewer #2: No

Reviewer #3: No

---

## [Author Response · Author response to Decision Letter 0]

30 Dec 2022

Dear Editor, Tommaso Martino, and my 3 reviewers,

Thank you very much for your kind reviews. Your comments are very useful for us to improve our manuscript. Thank you for give us an opportunity to correct many points in manuscripts for improve the quality.

---

## [Decision Letter · Decision Letter 1]

2 Feb 2023

PONE-D-22-27465R1Health-related quality of life of daily-life-affected benign essential blepharospasm: multi-center observational studyPLOS ONE

Dear Dr. Hirunwiwatkul,

Thank you for submitting your manuscript to PLOS ONE. After careful consideration, we feel that it has merit but does not fully meet PLOS ONE’s publication criteria as it currently stands. Therefore, we invite you to submit a revised version of the manuscript that addresses the points raised during the review process.

ACADEMIC EDITOR: I received positive evaluation from our eminent referees. There are still some minor comments that needs to be addressed (see below). A further revision of your english writing style is needed. 

We look forward to receiving your revised manuscript.

Kind regards,

Tommaso Martino, M.D.

Academic Editor

PLOS ONE

**Journal Requirements:**

****

**Additional Academic Editor Comments:**

In addition to the comments of our eminent referees, please consider the following:

- Line 63: add p-values.

- Line 77: change "depends of multifactor" with "are multifactorial".

- Line 78: change "another" with "one of".

- Line 86: change "that" with "whose".

- Line 108: remove the sentence from "we would like..." to "... then the". Start with inclusion and exclusion criteria, which should be reported in a more systematic way.

- Line 121: change "was pregnant" with "were pregnant".

- Line 125: explain what is the most common "standard routine" used in your centers.

- Line 178 and 185: change "underlying diseases" with "comorbid diseases" or "comorbidity".

- Line 188: I would not say "significant number" if the percentage is 12.5%.

- Line 190: what do you mean by "transportation cost"? And why is it reported?

- Line 196-198: acronyms should be listed in alphabetical order.

- Line 203-205: why are you reporting the predicted scores using the linear mixed model? What is the meaning to give to the predicted scores, compared to the actual scores? Similarly for the other reported scores (line 219, etc.)

- Line 214: remove the opening parenthesis.

- Line 257-262: move the paragraph from "a recent multicenter..." to "... and specificity [27]" in the limitations section of the article.

- Line 262-291: move the sentences after the discussion of your results. The discussion section should be focused on the results of YOUR paper, and then a comparison with previous literature.

- Line 318: start a new paragraph when reporting the strength and limitations of your paper. Another limitation is the fact that some interview (how many?) were performed by remote, due to COVID19 pandemia.

- In the introduction it is said that the treatment with BTX-A is not covered by your health system. How do you treated the patients included in your study? Do they payed the treatment? This could also be a selection bias, that should be included in the limitations section.

Tommaso Martino, M.D.

Academic Editor

PLOS ONE

Reviewers' comments:

Reviewer's Responses to Questions

**Comments to the Author**

1. If the authors have adequately addressed your comments raised in a previous round of review and you feel that this manuscript is now acceptable for publication, you may indicate that here to bypass the “Comments to the Author” section, enter your conflict of interest statement in the “Confidential to Editor” section, and submit your "Accept" recommendation.

Reviewer #1: All comments have been addressed

Reviewer #2: All comments have been addressed

Reviewer #3: All comments have been addressed

2. Is the manuscript technically sound, and do the data support the conclusions?

Reviewer #1: Yes

Reviewer #2: Yes

Reviewer #3: Yes

3. Has the statistical analysis been performed appropriately and rigorously? 

Reviewer #1: Yes

Reviewer #2: Yes

Reviewer #3: Yes

4. Have the authors made all data underlying the findings in their manuscript fully available?

Reviewer #1: No

Reviewer #2: Yes

Reviewer #3: Yes

5. Is the manuscript presented in an intelligible fashion and written in standard English?

Reviewer #1: Yes

Reviewer #2: Yes

Reviewer #3: Yes

6. Review Comments to the Author

Reviewer #1: The authors have addressed all the points, and they have strengthen the manuscript.

Reviewer #2: The authors aswered to most of the comments. However, I have further recommendations in order to get acceptable english language. Please check the following sentences:

1. line 86: “Blepharospasm is one of focal dystonias that symptoms commonly spread to…”

2. lines 114-115 : “We use the JRS, worldwide for clinical research and easy to use [21], for assess the severity and frequency of the blepharospasm”

3. line 288 : “Male and female patients might aware of the…”

4. lines 301-302 : “Mental health and difficulties with vision activities in near, distance and peripheral had effects…”

5. line 316: “BTXA treatment is useful for decreased physical severity…”

6. lines 320-322 : “The large 320 network of cooperation from participating neuro321

ophthalmologists and neuro-medicine units to deploy both generic and visual-specific

322 questionnaires allowed for broader data collection however, this study did have some limitations”

Moreover, Martino et al., 2012 and Weiss et al.,2006 references are reported twice in refercences list. (9-30 and 8 – 29 , respectively); Yang et al., 2021 reference is reported three times (5-7-12);

Reviewer #3: I did non ask for revision of the previous version, the revised version is still accurate and I can confirm that the current version of the manuscript is worthy of publication.

7. PLOS authors have the option to publish the peer review history of their article (what does this mean?). If published, this will include your full peer review and any attached files.

Reviewer #1: No

Reviewer #2: No

Reviewer #3: No

---

## [Author Response · Author response to Decision Letter 1]

19 Feb 2023

Dear Editors and Reviewers,

Thank you for your kind comments. I got many points of view and grammar to improve my manuscript. Here is my explanations and additional contents which already added in the 2nd revised manuscript. 

Additional Academic Editor Comments:

In addition to the comments of our eminent referees, please consider the following:

- Line 63: add p-values.

>> Done (p<0.001)

- Line 77: change "depends of multifactor" with "are multifactorial".

>> Done

- Line 78: change "another" with "one of".

 >> Done

- Line 86: change "that" with "whose".

 >> Done 

- Line 108: remove the sentence from "we would like..." to "... then the". Start with inclusion and exclusion criteria, which should be reported in a more systematic way.

 >> Done

- Line 121: change "was pregnant" with "were pregnant".

 >> Done

- Line 125: explain what is the most common "standard routine" used in your centers.

 >> Done. Change to “All participants received BTX-A treatment from each center to control the spasm, and the dose depended on the severity, previous response and type of BTX-A.”

- Line 178 and 185: change "underlying diseases" with "comorbid diseases" or "comorbidity".

 >> Done

- Line 188: I would not say "significant number" if the percentage is 12.5%.

 >> Done. 

- Line 190: what do you mean by "transportation cost"? And why is it reported?

 >> In this project, we plan to use the information from this study to do the economic analysis in the future, then we collect cost of transportation from the patients. In this article, we showed about QOL in patients with daily-life affected blepharospasm and improvement after Botulinum toxin treatment. Then, we removed the cost of transportation from this article, in results and in the table 1.

- Line 196-198: acronyms should be listed in alphabetical order.

 >> Done

- Line 203-205: why are you reporting the predicted scores using the linear mixed model? What is the meaning to give to the predicted scores, compared to the actual scores? Similarly for the other reported scores (line 219, etc.)

 >> We did not aim to compare the predicted score to the actual scores. Since the quality-of-life scores were measured repeatedly in three visits within each patient. Thus, the data for each patient is correlated. To adjusted for correlated data, we use a linear mixed model to estimate the scores and we called the scores estimated from the linear mixed model as a predicted score. We just would like to show the value of scores after adjusting for the correlated nature of the data.

- Line 214: remove the opening parenthesis.

 >> Done

- Line 257-262: move the paragraph from "a recent multicenter..." to "... and specificity [27]" in the limitations section of the article.

 >> Already moved and rearranged to line 315-321

- Line 262-291: move the sentences after the discussion of your results. The discussion section should be focused on the results of YOUR paper, and then a comparison with previous literature.

 >> Already rearranged in line 258-279 

- Line 318: start a new paragraph when reporting the strength and limitations of your paper. Another limitation is the fact that some interview (how many?) were performed by remote, due to COVID19 pandemia.

 >> Already restarted paragraph of strength and limitations. Also rewrite in some part of the limitation 315-330

- In the introduction it is said that the treatment with BTX-A is not covered by your health system. How do you treated the patients included in your study? Do they payed the treatment? This could also be a selection bias, that should be included in the limitations section.

 >> Done. Already added in limitation. “The new patients who self-pay their medical expenses might deny Botulinum toxin treatment due to financial limitation. They, thus, were not including into the study. This resulted in another potential selection bias.” Line 322-4

Reviewer #2: 

Reviewer #2: The authors aswered to most of the comments. However, I have further recommendations in order to get acceptable english language. Please check the following sentences:

1. line 86: “Blepharospasm is one of focal dystonias that symptoms commonly spread to…”

 >> Done. Blepharospasm is one of focal dystonias whose symptoms commonly spread to… as editor suggested.

2. lines 114-115 : “We use the JRS, worldwide for clinical research and easy to use [21], for assess the severity and frequency of the blepharospasm”

 >> Done. We use the JRS, which is widely used for clinical research and is easy to use [21], to assess the severity and frequency of blepharospasm.

3. line 288 : “Male and female patients might aware of the…”

 >> Done. Male and female patients may perceive the affected quality of life differently, even if the severity is the same. Line 280-281

4. lines 301-302 : “Mental health and difficulties with vision activities in near, distance and peripheral had effects…”

 >> Done. From NEI-VFQ25, mental health, difficulty with near-vision activities, distance-vision activities and peripheral vision had effects in the 50-60% range. Line 293-4

5. line 316: “BTXA treatment is useful for decreased physical severity…”

 >> Done: BTX-A treatment not only reduces physical severity but also enhances quality of life, including in non-physical aspects. Line 308-9

6. lines 320-322 : “The large network of cooperation from participating neuro ophthalmologists and neuro-medicine units to deploy both generic and visual-specific questionnaires allowed for broader data collection however, this study did have some limitations”

 >> Done. The strength of this study is that we conducted a prospective multicenter, observational study from neuro-ophthalmologists and neurologists in 14 centers nationwide with large population that included participants from all parts of Thailand. Data on quality of life, collected through generic and visual-specific questionnaires, broadened the scope of data collection. Line 311-4

Moreover, Martino et al., 2012 and Weiss et al.,2006 references are reported twice in refercences list. (9-30 and 8 – 29 , respectively); Yang et al., 2021 reference is reported three times (5-7-12);

 >> Thank you very much for your information. I already revised all the references. 

Response for all reviewers: 

Thank you very much for your kind reviews. Your comments are very useful for us to improve our manuscript.

---

## [Decision Letter · Decision Letter 2]

2 Mar 2023

Health-related quality of life of daily-life-affected benign essential blepharospasm: multi-center observational study

PONE-D-22-27465R2

Dear Dr. Hirunwiwatkul,

We’re pleased to inform you that your manuscript has been judged scientifically suitable for publication and will be formally accepted for publication once it meets all outstanding technical requirements.

Kind regards,

Tommaso Martino, M.D.

Academic Editor

PLOS ONE

Reviewers' comments:

Reviewer's Responses to Questions

**Comments to the Author**

1. If the authors have adequately addressed your comments raised in a previous round of review and you feel that this manuscript is now acceptable for publication, you may indicate that here to bypass the “Comments to the Author” section, enter your conflict of interest statement in the “Confidential to Editor” section, and submit your "Accept" recommendation.

Reviewer #1: All comments have been addressed

Reviewer #2: All comments have been addressed

2. Is the manuscript technically sound, and do the data support the conclusions?

Reviewer #1: Yes

Reviewer #2: Yes

3. Has the statistical analysis been performed appropriately and rigorously? 

Reviewer #1: Yes

Reviewer #2: Yes

4. Have the authors made all data underlying the findings in their manuscript fully available?

Reviewer #1: No

Reviewer #2: Yes

5. Is the manuscript presented in an intelligible fashion and written in standard English?

Reviewer #1: Yes

Reviewer #2: Yes

6. Review Comments to the Author

Reviewer #1: I have no further comments to this manuscript.

Reviewer #2: (No Response)

7. PLOS authors have the option to publish the peer review history of their article (what does this mean?). If published, this will include your full peer review and any attached files.

Reviewer #1: No

Reviewer #2: No

---

## [Editor Report · Acceptance letter]

6 Mar 2023

PONE-D-22-27465R2 

Health-related quality of life of daily-life-affected benign essential blepharospasm: multi-center observational study 

Dear Dr. Hirunwiwatkul:

I'm pleased to inform you that your manuscript has been deemed suitable for publication in PLOS ONE. Congratulations! Your manuscript is now with our production department. 

Kind regards, 

on behalf of

Dr. Tommaso Martino 

Academic Editor

PLOS ONE